

# Vertical characteristics of aerosol hygroscopicity and impacts on optical properties over the North China Plain during winter

**Quan Liu[1,3], Dantong Liu[2*], Qian Gao[1,4], Ping Tian[4,5], Fei Wang[1], Delong Zhao[1], Kai Bi[1], Yangzhou Wu[2], Shuo Ding[2], Kang Hu[2], Jiale Zhang[2], Deping Ding[1], Chunsheng Zhao[6]**

[1] Beijing Weather Modification Office, Beijing 100089, China
[2] Department of Atmospheric Sciences, School of Earth Sciences, Zhejiang University, Hangzhou,
Zhejiang, 310027, China
[2] Institute of Urban Meteorology, Chinese Meteorological Administration, Beijing 100089, China
[4] Beijing Key Laboratory of Cloud, Precipitation and Atmospheric Water Resources, Beijing, 100089, China
[5] Field experiment base of cloud and precipitation research in North China, China Meteorological
Administration, Beijing, 101200, China
[6] Department of Atmospheric and Oceanic Sciences, School of Physics, Peking University, Beijing 100871, China

Corresponding author: Dantong Liu (dantongliu@zju.edu.cn)



## Abstract

The water-uptake on aerosol influences its optical depth and capacity of cloud formation, depending on the vertical profile of aerosol hygroscopicity because of different solar radiation received and
supersaturation conditions at different atmospheric levels. Such information is lack over the polluted eastern Asian region. This study presents aircraft in-situ measured aerosol size distribution and chemical compositions by series of flights over Beijing area in wintertime. Under high relative humidity conditions (surface RH>60%, hRH), a significant enhancement of aerosol hygroscopicity parameter ($\kappa$) in the planetary boundary layer (PBL) was observed to increase by 50% up to 0.32 from the surface to the top
of PBL (vertical gradient of ~0.13 km$^{-1}$), along with the dry particle effective diameter ($D_{eff}$) increased by 71% and activation ratio up to 0.23 (0.64) at SS=0.05% (0.1%), in contrast with a lower vertical gradient of $\kappa$ (0.05 km$^{-1}$) and smaller $D_{eff}$ under low RH conditions (surface RH<60%, lRH). This suggests the aqueous processes played an important role on promoting the enhancement of particle hygroscopicity in the PBL. The $\kappa$ in the lower free troposphere (LFT) was relatively stable at 0.24±0.03 with slight increase
during regional transport. The enhancement of aerosol optical depth (AOD) due to water uptake ranged at 1.0-1.22 for PBL under lRH and LFT, but reached as high as 6.4 in the PBL under hRH. About 80% and 18% of the AOD was contributed by aerosol hygroscopic growth under hRH and lRH respectively. These results emphasize the importance of boundary layer processing on aerosol water-uptake capacity especially under high RH condition.



## 1 Introduction

The water growth on particle could increase particle size and modify its refractive index hereby affecting its radiative effects. The aerosol can be subject to hygroscopic growth under sub-saturation (Köhler, 1936) and serve as cloud condensation nuclei (CCN) under supersaturated environment (Dusek et al., 2006). In addition, as most instruments characterize the properties of aerosol in dry condition, it is necessary to recover the properties to the ambient environment when using the observational data to estimate the direct and indirect radiative impacts of aerosol.

The hygroscopic properties of aerosol are mainly determined by composition, with inorganics having higher hygroscopicity (Cruz and Pandis, 2000; Gysel et al., 2007) than less water-soluble substances, such as black carbon (Aklilu et al., 2006; Pringle et al., 2010) or primary organics (Wang et al., 2008). However, ambient aerosols are complex mixtures and their compositions vary at different stages of atmospheric ageing process (Zhang et al., 2007). A single hygroscopicity parameter($\kappa$) (Petters and Kreidenweis, 2007) is used to describe the composition effect on hygroscopicity, under both sub- and super-saturation condition (Petters and Kreidenweis, 2008). The aerosol composition measurements were intensively conducted on the ground at East Asia region in recent decade (Meng et al., 2014; Wu et al., 2016; Zou et al., 2019; Irwin et al., 2011). These studies computed the measured compositions by volume-weighted fractions to estimate the $\kappa$, and found $\kappa$ ranging from 0.1 - 0.4 under different environments or pollution sources, in particular, the secondary inorganics is in consensus found to be the main component driving the liquid water content in aerosol (Pringle et al., 2010; Prenni, 2003; Khlystov, 2005). The water absorbed on aerosol could importantly influence the consequent gas uptake (Kolb et al., 2002) and aqueous reactions (Ge et al., 2012), and may further promote secondary formation in particle phase (Hennigan et al., 2008).

The boundary layer meteorology and associated physiochemical processing on pollutants has raised great attention recently, which could cause important feedback impacts on enhancing the pollution level via



inhibiting the development of boundary layer (Zhou et al., 2019; Bharali et al., 2019; Liu et al., 2018b).

This impact is importantly determined by vertical distribution of aerosol concentration, size distribution, and optical properties. The location of aerosol layer, or hygroscopic growth at different locations in the atmosphere column is important in altering the thermodynamic stability, e.g. on influencing the radiative inversion through dimming effect towards lower level (T. Morgan et al., 2010; Massoli et al., 2009). Under

high pollution, this impact could be exacerbated, especially under high moisture condition, as evidenced by a number of studies that over 25% of the polluted days with significantly reduced visibility in megacities were associated with high RH (Deng et al., 2013; Zhong et al., 2017; Qiang et al., 2015; Quan et al., 2014; Liu et al., 2013b). These emphasize the importance in studying the vertical characteristics of particle hygroscopicity, but such information is still lack due to limited airborne measurements over

eastern Asia region.

This study reports the results of series of aircraft in-situ measurements conducted over Beijing region in 2016 winter. The detailed chemical compositions are used to estimate the vertical distributions of aerosol hygroscopicity. The in-situ measured size distribution and hygroscopic growth factor are combined to evaluate the influence of water uptake on the ambient aerosol optical depth (AOD) and CCN activation

ratio under different moisture conditions.

## 2 Experimental and data analysis

2.1 Flight information

Aircraft measurements were performed over Beijing area by the KingAir-350 aircraft in 2016 winter (Liu

et al., 2018a; Tian et al., 2019; Zhao et al., 2019). The sampling inlet system used on the aircraft is the Model 1200 passive Isokinetic Aerosol Sampling Inlet (BMI, Brechtel Manufacturing Inc), which could deliver 150 lpm of sample flow at 100 m s$^{-1}$ air speed, with particle diameters between 0.01-6μm with >95% collection efficiency (Tian et al., 2019; Hermann et al., 2001). The maintained room temperature in the





cabin serves as an automatic drier when ambient temperature was lower than inside at higher altitude; in
addition to that, a silicate diffusion drier was installed before the instrument sampling, further warranting
the dry condition for the samples. An Aircraft Integrated Meteorological Measurement System (AIMMS-20, Aventech Inc., Canada) is mounted under the wing to measure temperature (T), relative humidity (RH), wind speed/direction and pressure with a time resolution of 1s.

The operation of flights was carried out to avoid clouds where possible, and the results here have been
screened to remove the in-cloud data, as determined by measurements of relative humidity and cloud liquid water content. The flights were mostly operated at altitudes up to 2.5km, focusing on the pollutants in the planetary boundary layer (PBL) and lower free troposphere (LFT) around Beijing area. The flight tracks and time schedules are presented in Figure 1 and Table 1. The aircraft took off from Shahe in the morning (a rural area ~20km to the north-west of central Beijing), conducting a full profile, and then
flying over the Beijing city or the surrounding area with a few constant-level runs at different altitudes, at last followed by another full profiling over Shahe. As the aircraft was unable to fly lower than 500m altitude over Beijing city due to flight-path restrictions, thus full profiles throughout the lower troposphere were only conducted over Shahe.

2.2 Instrumentation for aerosol measurements

The particle size distribution was measured by the PCASP (passive cavity aerosol spectrometer probe) instrument with a time resolution of 1 s, at diameters from 0.1 to 3 μm. With a wired heater on top of the inlet, the aerosol size distribution measured by the PCASP was considered to be dry at RH < 40% (Walter Strapp et al., 1992). Due to the detection limit of the instrument, the first two bins (0.1-0.11 μm, 0.11-0.12 μm) are eliminated from the analysis (Liu et al., 2009). The aerosol number concentration Na (cm$^{-3}$)
refers to the total number concentration with diameter 0.12 - 3 μm. The effective diameter $D_{eff}$ is calculated by:

$$D_{eff} = \sum_i N_i D_i^3 / \sum_i N_i D_i^2 \quad (1)$$



$N_i$ is the number concentration of $i^{th}$ size bin; $D_i$ is the particle diameter at each size bin.

A Compact time-of-flight aerosol mass spectrometer (C-ToF-AMS) measured submicron non-refractory aerosol (NR-PM$_1$) chemical compositions with time resolution of 1 minute, including nitrate (NO3), sulphate (SO4), ammonium (NH4), chloride (Chl) and organics (Org) (Drewnick et al., 2005; Canagaratna et al., 2007). The term non-refractory refers to all species that can be vaporized at 600 °C and $\sim 10^{-7}$ Torr. A constant pressure controller was used to regulate and maintain the downstream pressure at 650 hPa, in order to ensure constant sampling conditions for the AMS during altitude change (Bahreini et al., 2008). All calibrations (flowrate, particle velocity, ionization efficiency) were performed under this pressure before and after each flight. Mass concentrations derived from the AMS are reported as micrograms per standard cubic metre ($T$=273.15 K, $p$=1013.25hPa). The AMS collection efficiency (CE), which accounts for the incomplete detection due to particle bounce at the vaporiser and/or the partial transmission of particles by the lens (Canagaratna et al., 2007), is significantly modulated by particle phase (Matthew et al., 2008). In this study, a CE correction was used following Middlebrook et al. (2012). A Positive Matrix Factorisation (PMF) analysis was performed on the organic mass spectra following the procedures by Ulbrich et al. (2009). Two factors were resolved for the results here, which are the hydrocarbon-like organic aerosol (HOA) and oxygenated organic aerosol (OOA), corresponding to the primary OA (POA) and secondary OA (SOA), respectively.

Equivalent black carbon mass was measured with an aethalometer (AE33, MAGEE Scientific) at 1 Hz. The aethalometer collected aerosol particles through the same isokinetic inlet and sampling line as for the AMS. The instruments used dual-dots configuration to auto-correct for the loading affect. The measured absorption was converted to BC mass using an apparent mass absorption cross section (MAC) of 7.7 m$^2$ g$^{-1}$ at a wavelength of 880 nm (Drinovec et al., 2015). The $\lambda$=880nm is chosen to avoid the potential interference of brown carbon at shorter wavelength. The multi-scattering enhancement factor (C value) of 2.88 at 880nm wavelength was used to exclude the multiple light scattering effects, which was obtained





through a laboratory study by running the AE33 in parallel with a photoacoustic photometer (PASS-3, DMT, USA) for one week ambient measurement (Tian et al., 2019).

The measurements of the AMS and aethalometer, which are the non-refractory and refractory composition respectively, represent the main compositions of aerosol in PM$_1$. The sum of AMS and AE33 measured mass is compared with PCASP-derived PM$_1$ (Figure S1), and showed high correlation ($R^2$=0.91, slope=1.05), implying the high agreement of measurements between inside and outside the cabin.

## 2.3 Aerosol hygroscopic properties

The hygroscopic parameter $\kappa$ (Petters and Kreidenweis, 2007) is solely determined by composition and
reflects the Raoult term in Köhler theory. The $\kappa$ for an internal mixture with multiple compositions is contributed by $\kappa$ of each volume-weighted composition, following the Zdanovskii–Stokes–Robinson (ZSR) mixing rule (Stokes and Robinson, 1966), expresses as:

$$\kappa = \sum_i \varepsilon_i \kappa_i \quad (2)$$

where $i$ represents the $i^{th}$ composition, $\varepsilon_i$ is the volume fraction of each composition in the bulk, and $k_i$ is
the hygroscopic parameter for each composition. In this study, the compositions are determined by AMS the AE33 measurements. In particular, the inorganic compositions are derived by empirically pairing the AMS-measured ions (Gysel et al., 2007), expressed as:

$$
\begin{aligned}
n_{NH_4NO_3} &= n_{NO_3^-} \\
n_{H_2SO_4} &= \max\left(0, n_{SO_4^{2-}} - n_{NH_4^+} + n_{NO_3^-}\right) \\
n_{NH_4HSO_4} &= \min\left(2n_{SO_4^{2-}} - n_{NH_4^+} + n_{NO_3^-}, n_{NH_4^+} - n_{NO_3^-}\right) \quad (3), \\
n_{(NH_4)_2SO_4} &= \max\left(0, n_{NH_4^+} - n_{NO_3^-}\right) \\
n_{HNO_3} &= 0
\end{aligned}
$$

All species are then converted to volume by assuming a density. Table 2 summaries the density and $\kappa$ used
for all species mentioned in this study. The $\kappa$ of organics ($\kappa_{org}$) has more diversity compared to inorganics (Saxena et al., 1995; Aklilu et al., 2006). Previous studies suggest that the hygroscopicity of organics varied with their oxidation state (Chang, 2011; Tritscher et al., 2011). The organic matter was classified


as primary organic aerosol (POA)POA and secondary organic aerosol (SOA) by the PMF analysis. According to a closure study between aerosol chemical composition and hygroscopic growth in Beijing

(Wu et al., 2016), the hygroscopicity of organic matter was assigned with a $\kappa_{SOA}$ and $\kappa_{POA}$ of 0.1 and 0 respectively, and $\kappa_{BC}$ is set to 0.

2.4 Aerosol optical properties

The refractive index (RI) in bulk as contributed by different compositions is calculated according to the volume mixing rule (Wen, 2003). The RI of each volume-weighted composition is summarized in Table

2. In addition to dry compositions, the volume of water contained in particle is calculated based on the hygroscopic growth of particle under certain RH. If the hygroscopicity parameter ($\kappa$) is known, aerosol hygroscopic growth factor (HGF) and ambient size distribution can be calculated from the dry particle diameter ($D_d$) and ambient relative humidity (RH), expressed as:

$$\frac{RH}{\exp\left(\frac{A}{D_d HGF}\right)} = \frac{HGF^3 - 1}{HGF^3 - (1 - \kappa)} \quad (4)$$

$$A = \frac{4\sigma_{s/a} M_w}{RT\rho_w} \quad (5)$$

where $\sigma_{s/a}$ is the water surface tension at the solution-air interface, $M_w$ is the molar mass of water, $R$ is the universal gas constant, $T$ is the absolute temperature and $\rho_w$ is the density of water.

The volume of absorbed water ($V_{water}$) is then calculated from $HGF$ by:

$$V_{water} = \frac{\pi}{6} D_d^3 (HGF^3 - 1) \quad (6)$$

The water is then taken into account as a composition to work out the RI for wet aerosol, expressed as:

$$m_{amb} = \sum_i \left(\frac{V_i}{V_{chem} + V_{water}}\right) m_i \quad (7)$$

$$n_{amb} = \sum_i \left(\frac{V_i}{V_{chem} + V_{water}}\right) n_i \quad (8)$$

where $V_i$ is the respective volume of each component, $V_{chem}$ and $V_{water}$ is total volume of all chemical



species (other than water) and absorbed water respectively; $m_i$ and $n_i$ are real and imaginary parts of refractive indices for each pure component. Then the real part ($m_{amb}$) and imaginary part ($n_{amb}$) of ambient aerosol particle refractive indices can be derived from chemical components and absorbed water by equation (7) and (8).

The extinction cross section ($C_{ext}$, in μm$^2$) is calculated at each particle diameter ($D_i$), multiplied by number centration (in cm$^{-3}$) at each $D_i$ to obtain the extinction coefficient ($\sigma_{ext}(D_i)$, in Mm$^{-1}$). The $\sigma_{ext}(D_i)$ is then integrated over all $D_i$ distribution to obtain the total $\sigma_{ext}$ for bulk aerosol at specified wavelength (λ, 800nm in this study). This calculation is performed for both dry and ambient conditions using dry and wet particles size, particle RI (as calculated above) to obtain the dry or ambient total extinction. The $\sigma_{ext}$ is multiplied by height interval (Δh, 100m) to obtain the dry and ambient aerosol optical depth (AOD) at each altitude:

$$AOD_{dry}(h, \lambda) = \Delta h \times \sum_i C_{ext,dry}(D_{i,dry}, \lambda)N_i \qquad (9)$$

$$AOD_{amb}(h, \lambda) = \Delta h \times \sum_i C_{ext,amb}(D_{i,amb}, \lambda)N_i \qquad (10)$$

$D_{i,dry}$ is the dry particle diameter, $D_{i,amb}$ is calculated by $D_{i,dry}$ multiplied by HGF, which represents the ambient particle diameter under ambient RH condition.

$f$(AOD), which is the ratio of AOD$_{amb,100m}$ and AOD$_{dry,100m}$, is introduced to characterize the AOD enhancement due to particle hygroscopic growth under ambient condition.

## 3 Results and discussions

### 3.1 Meteorology

Vertical profiles of aircraft in-situ measured meteorological parameters (temperature T, potential temperature $\theta$, relative humidity RH, water mixing ratio $q$) under high and low relative humidity





conditions are presented in Figure 2. The high and low RH conditions are defined by ground-level RH higher and smaller than 60%, respectively. The height of planetary boundary layer (PBL) is defined as the altitude (z) at which the vertical gradient $d\theta/dz$ reached 10K/km, and in the PBL $d\theta/dz$ less than 10 K km$^{-1}$ denoted a thermal-dynamically well mixed layer (Su et al., 2017). As shown in Figure 2a,

temperature inversion layers appeared on top of the PBL for most flights, and the degree of inversion under high RH condition was much larger than that under low RH condition, with mean values of 7.8°C. Along with temperature decrease in vertical direction, RH in the PBL showed positive vertical gradient in the PBL, especially under high RH condition (Figure 2c). The water mixing ratio ($q$) showed weak vertical variation in the PBL (Figure 2d and h), meaning a well-mixed moisture.

Recent study found in Beijing most aerosols deliquesced at RH~60% (Zou et al., 2019), a criteria with surface RH above or below 60% is thus set to investigate the potential moisture influence on the observed composition, defined as high RH (hRH) and low RH (lRH) respectively. For lRH cases, the profiles were further classified as more polluted condition when surface PM$_1$ > 100 μg m$^{-3}$.

3.2 Vertical characterization of aerosol chemical composition

The vertical profiles of aerosol chemical components under lRH and hRH conditions are shown in Figure 3, including the primary emissions (BC, chloride, POA), secondary compositions (nitrate, sulfate, SOA), and the ratio between both, i.e. SOA/POA, SPM/BC (SPM is the sum of secondary species). Because the primary sources mainly result from surface emission, all primary species (BC, Chl, POA) featured with

an accumulated concentration towards lower level, but a reduced concentration at higher level. This consistent exponential decrease profile pattern in wintertime was also observed in previous studies over Beijing (Zhang et al., 2009; Liu et al., 2009; Zhao et al., 2019). However, the mass concentrations for all secondary components including nitrate, sulphate and SOA had less vertical gradient within the PBL (Figure 3e-h). This is further reflected by Figure 3i-j, with the secondary/primary ratio (SOA/POA,



SPM/BC) showing pronounced positive vertical gradient and this increase was capped on top of the PBL. It is noted that the increased contribution of secondary species was closely corelated with RH increase in the PBL. The increased RH could promote the condensation of semi-volatile species to the aerosol phase (Khlystov et al., 2005; Pankow et al., 1993) and may also enhance the heterogeneous reactions on the existing particle surface from gaseous precursors (Guo et al., 2014; Huang et al., 2014). Due to the higher

hygroscopicity of secondary species, the observations here provide direct evidence that the increase of moisture had modified the aerosol composition in the PBL to contain more secondary species and more hygroscopic.

For the lRH condition (surface RH<60%), contrasting vertical structures of aerosol compositions were observed compared to hRH. The aerosol loadings had large variabilities, and the high concentration in the

PBL coincided with the reduced the PBL height. These conditions are thus further classified as more and less polluted corresponding with PBL height <500 m and >500 m respectively. The secondary species almost covaried with the primary, leading to an almost consistent secondary/primary ratio in the PBL, with SOA/POA ~2.1 and SPM/BC ~9.5 (Figure 3i-j). Under polluted condition both ratios were lower than that under less polluted condition. The contribution of secondary aerosols as reflected by SOA/POA

and SPM/BC, fell within the same range with that at the surface level of hRH. By comparing with the hRH condition, the almost maintained secondary contribution in the PBL under lRH (Figure 3i-j) suggests the less important secondary formation, or at least the moisture in the PBL had not sufficiently promoted the modification of primary species, but the pollutants were mainly modulated by the emissions and regional transport.

Figure 3h showed $PM_1$ mass concentrations showed exponential decreases with altitude in the LFT with most concentrations distributed in the range of 2-38 μg m$^{-3}$, and the contribution of SOA became more significant (Figure S2).



### 3.3 Vertical profile of particle hygroscopicity

Figure 4 shows the vertical profiles of hygroscopicity parameter ($\kappa$) and effective diameter under all conditions. The bulk $\kappa$ is largely modulated by secondary inorganic compositions given their larger $\kappa$. The $\kappa$ on the ground showed consistent 0.22±0.02 (range of 0.20-0.25) under all conditions, which was in the middle range of previous ground measurements in Beijing (Wang and Chen, 2019; Wu et al., 2016; Zou et al., 2019; Liu et al., 2013a), and the observations here extend the hygroscopicity information to the

upper level. As shown in Figure 4a, the vertical profiles of $\kappa$ under hRH show a pronounced increase from surface level to the top of PBL with a variation from 0.18 to 0.34 by a factor of 1.9. This is consistent with the increasing fraction of the most secondary inorganic hygroscopic species due to pure inorganic substance is more hygroscopic (Table 2). The increase of $\kappa$ generally followed a linear correlation with a slope of 0.13 km$^{-1}$, and in contrast with a much lower vertical gradient of $\kappa$ (slope=0.05 km$^{-1}$) under lRH.

Under hRH, the source of moisture from the surface was accumulated in the PBL and promoted the enhancement of particle hygroscopicity thus showing a positive correlation between $\kappa$ and RH (Figure 2c and Figure 4a). This means under hRH condition the aerosol in the PBL significantly enhanced the capacity of water uptake and deliquesce process in vertical direction, thus provides a more reactive surface for aerosol to enhance the condensation and aqueous reaction.

The $\kappa$ showed maxima at the top of PBL under hRH and lRH less polluted conditions, then in the LFT the secondary fraction and $\kappa$ decreased with altitude. $\kappa$ showed higher value above the PBL under lRH polluted condition compared to the others at the same height. Back-trajectory analysis (Figure S5) showed that these aerosols advected by regional transport from the polluted southern region (Liu et al., 2018a; Tian et al., 2019) may have already been aged and hygroscopic. The $D_{eff}$ showed large variations in the

PBL and depended on the pollution level and RH. In line with the $\kappa$, high RH condition also showed a remarkable enhancement of $D_{eff}$ from the surface to the top of the PBL by 71%, while under lRH the $D_{eff}$ had almost no vertical variation and larger $D_{eff}$ showed under higher pollution. The $D_{eff}$ was consistently





at 0.25-0.32 µm in the LFT for all conditions (Figure 4b).

Figure 4c,d summarizes the $\kappa$ and $D_{eff}$ in the PBL and LFT under the three types of environments. In the

PBL, $\kappa$ showed consistency at 0.24±0.02 under different pollution levels of lRH but $D_{eff}$ variaed at 0.28-

0.38 µm respectively. However, the notable increase of both $\kappa$ and $D_{eff}$ under hRH suggests the important

aqueous processes on modifying both the particle size and chemical compositions (Qiang et al., 2015;

Sun et al., 2016), particularly at the top of the PBL (Liu et al., 2018b). The particles in the LFT, showing

$\kappa$ at 0.23-0.26 but consistently smaller particle size at $D_{eff}$ =0.27-0.30 µm, which may result from a lack

of gas-precursors at upper level not allowing particle growth.

3.4 Dry and ambient size distribution

Combing the measured size distribution and hygroscopicity information, the aerosol size distribution

under both dry and ambient conditions can be obtained. Figure 5 shows the typical examples of aerosol

dry and ambient size distribution under different conditions. The hygroscopic growth factor (HGF) of

particle in the ambient is determined by RH and hygroscopic parameter $\kappa$. As Figure S3 shows, the HGF

exponentially increased with ambient RH, and at higher $\kappa$ this increase had a higher offset. When

RH<60%, the HGF only slightly increased with RH, however, HGF exponentially increased with RH at

higher RH, e.g., from RH 80% to 95%, HGF increased from 1.2 to 2.1 by a factor of 1.8. Hygroscopicity

also exerts more significant impacts on HGF under hRH conditions. As discussed in section 3.3, hRH

condition has increased both particle dry size and particle hygroscopicity, whereby the hygroscopic

growth could further enlarge particle size under high RH. This is demonstrated in Figure 5d, where

remarkable growth of aerosol size occurred in the hRH PBL with mean HGF of ~1.6 (Figure 5d). The

mean HGF for lRH was at 1.04±0.02, thus showing little difference between dry and ambient size

distribution under lRH condition due to lack of moisture for hygroscopic growth (Figure 5a-c).



3.5 Vertical profiles of particle dry and ambient AOD

The aerosol optical depth is derived from the dry and ambient size distribution. Figure 6 shows vertical profiles of dry and ambient AOD with height interval of 100m (AOD$_{100m}$) under lRH and hRH conditions.

For lRH less polluted (lRH_lp) periods (Figure 6a), the AOD was less than 0.02 throughout the column and showed insignificant vertical gradient, with AOD in the PBL slightly higher than that in the LFT. The AOD for lRH polluted (lRH_p) period could reach up to 0.040 and 0.043 for dry and ambient respectively. Over 70% of the integrated AOD was concentrated within the shallow PBL, and AOD above the PBL exponentially decreased with altitude (Figure 6b), where the difference between dry and ambient was

larger than lRH_lp. Consistent with the variation of $\kappa$ and particle size, AOD under high RH condition showed remarkable enhancement close to the top of PBL (Figure 6c), with dry and ambient AOD reaching up to 0.25 and 1.07 respectively.

The $f$(AOD) is hereby defined as the ratio of AOD$_{100m}$ between ambient and dry condition, to reflect the influence of hygroscopic growth on particle extinction. The vertical profiles are shown in Figure 6d, with

the mean±σ in the PBL and LFT are shown in Figure 6e. The $f$(AOD) is found to range at 1.0-1.2 for PBL under lRH and LFT at all conditions, but could reach as high as 4.4±1.3 in the PBL under hRH. $f$(AOD) is determined by combined factors of aerosol size, hygroscopicity and RH. The RH increased with altitude in the PBL under hRH conditions and deceased above the PBL (Figure 2c). The moisture trapped in the PBL enhanced the secondary aerosol formation through heterogeneous/aqueous reactions, as reflected by

the enhanced fraction of secondary inorganic and secondary organic components (Figure 3j) hereby increased hygroscopicity from surface to the top of PBL (Figure 4a). This is also consistent with the dry particle size, shown as the correlation between D$_{eff}$ (in the dry condition) and RH under lRH and hRH in Figure S4. When RH<60%, the D$_{eff}$ has no obvious correlation with RH, but significantly increased with RH when RH>60%. This is in line with the increased contribution of secondary species under hRH

condition. Consistent with the RH profile, both the peak D$_{eff}$ and peak $\kappa$ appeared at the top of PBL (Figure



4a and b), but all decreased above the PBL (apart from for the polluted lRH profiles there was an elevated $\kappa$ at higher altitude). This vertical structure was caused by a combination of the convective mixing in the PBL and a capping effect by the temperature inversion on top of the PBL. The gas, particle and moisture were trapped in the PBL where intensive deliquesce process and heterogenous/aqueous reactions occurred,

enlarging particle size and increasing particle hygroscopicity. These processes further led to peak $f$(AOD) appearing at top of the PBL (Figure 6d). Further back-trajectory analysis showed that for the polluted lRH profiles (e.g. flight on Dec. 18th), the enhanced $\kappa$ at ~1km above the PBL was introduced by regional transport from the polluted southwest region. For these cases, the aged particles as well as the moisture were advected from outside of the Beijing area, and the ageing processes as described above tended to

occur in the pathway of transport rather than occurring at local scale.

A comparison between in-situ measurement constrained AOD and AERONET AOD is presented in the Figure 7. Under low RH condition, the in-situ dry AOD has a high correlation with AERONET AOD ($R^2$=0.94) but 35% lower. Including the particle hygroscopic growth improves the agreement between both methods by 21%. This suggests a 7-25% of column-integrated AOD may be contributed by water

growth on particle under ambient surface RH<60%. Note that when ambient surface RH>60%, due to dramatically enhanced AOD in addition to the low-level cloud formation, the passive AERONET measurement was not available, we therefore only estimate the impacts of hygroscopic growth on AOD from our in-situ measurements. As the sub-panel of Figure 7 shows, under hRH, the AOD had been enhanced by a factor of 3.7-6.6 due to water uptake.


3.6 Vertical profile of CCN activity

The critical diameter $D_c$ is the diameter above which the particles are considered to be activated at a specific supersaturation (SS). The mean $D_c$ is determined from the bulk $\kappa$ (Petters and Kreidenweis, 2007), expressed as:



$$\kappa = \frac{4A^3}{27D_c^3 \ln^2 S_c} \qquad (11)$$

where A is defined by equation (5), $S_c$ is the critical supersaturation.

The total aerosol number concentration ($D_p$=0.12-3 μm) measured by the PCASP is denoted as $N_{CN}$. The CCN number concentration ($N_{CCN}$) is determined by the sum of the number concentration for the particle size larger than $D_c$. Hereby the CCN activation fraction ($N_{CCN}/N_{CN}$) in the diameter range of 0.12-3μm

can be obtained at a given SS.

Previous studies estimated the SS for stratus clouds to be slightly less than 0.1% over polluted continental regions, and a higher SS (often exceeded 1%) for stratus clouds in clean air masses (Hudson and Noble, 2014; Hudson et al., 2010). The North China Plain is one of the most polluted areas in China (Huang et al., 2014; Zhang et al., 2015), we thus test the CCN activity here at SS=0.05% and 0.1% respectively.

Figure 8 shows that the $D_c$ in hRH PBL was smaller than that in lRH PBL due to increased $\kappa$, and the vertical gradient of $D_c$ under hRH condition was larger than that under lRH. $D_c$ showed a higher variability at SS=0.05% than at SS=0.1% (Figure 8a, d), ranging from 0.27-0.35μm (SS=0.05%) and 0.18-0.21μm (SS=0.1%) respectively. Corresponding with $\kappa$ profiles shown in Fig 4a, both hRH and lRH_lp profiles showed minimum $D_c$ (at SS=0.05%) on top of the PBL at 0.27μm and 0.32μm respectively (Figure 8a).

The lRH_p showed elevated $D_c$ minima at ~1km above the PBL. At upper level in the LFT, $D_c$ increased with altitude for all conditions.

The $N_{CCN}$ showed enhanced concentration in the PBL than that of LFT, but with different vertical structures at different SS (Figure 8b, e). This is in line with the CCN activation fraction that a positive vertical gradient of $N_{CCN}/N_{CN}$ for hRH condition peaking at top of the PBL was shown at SS=0.05%, but

for lRH_lp condition, the $N_{CCN}/N_{CN}$ or $N_{CCN}$ was more uniformly distributed in the PBL. The increase of SS enhanced the vertical gradient of $N_{CCN}/N_{CN}$ for lRH_lp. It is noted that at SS=0.05% the potential CCN activation fraction of dry aerosol at the top of PBL was highest for hRH (0.23±0.04) and higher than





lRH_p by 53%. The increase of SS up to 0.1% led to a lessened difference among conditions, with lRH_p and hRH being more comparable. This suggests that the particle composition or size-dependent CCN
activation ability will be more homogenously distributed at higher supersaturation condition.

At which level the particle will be activated depends on the actual SS at different cloud levels, but the results here show that the enhanced RH will increase both dry particle size and hygroscopicity through a variety of aqueous reactions and processes. The particles are thus expected to be significantly activated at a level closer to cloud base (or higher temperature) and at a much lower altitude (lowered condensation
level due to increased surface RH), which will further depress the boundary layer development, hereby trapping the aerosol, gases and moisture within a more limited atmospheric column. The aerosols at higher level, which showed a smaller size and lower hygroscopicity, would need higher SS to be activated, though these particles tend to be activated or incorporated into clouds likely by entrainment from cloud top or larger-scale cloud system. The results here show that the surface characteristics of dry aerosols may
not present the particles which initialize the cloud formation at top of the PBL. Therefore, the process during pollutants uplifting from the surface to the top of the PBL until the particle activation point, should be considered, e.g. the enhancement of particle size and hygroscopicity with altitude in the PBL.

## 4 Conclusions

The vertical profiles of aerosol hygroscopic properties over the North China Plain were investigated based on the aircraft in-situ measured aerosol chemical compositions. These profiles covered ambient conditions of higher surface RH (hRH, >60%), lower RH (lRH, <60%) with less and more polluted conditions. For hRH, a significant enhancement of hygroscopicity parameter ($\kappa$) in the PBL was observed to increase by a factor of 1.9 from the surface to the top of PBL (generally following a linear correlation with a slope of
0.13 km$^{-1}$) along with the dry particle effective diameter ($D_{eff}$) increase by a factor of 1.7, in contrast with a much lower vertical gradient of $\kappa$ (slope=0.05 km$^{-1}$) and $D_{eff}$ under lRH. This suggests the aqueous





reaction played an important role on promoting the enhancement of particle hygroscopicity in hRH PBL. The $\kappa$ in the LFT was relatively stable at 0.24±0.02 with slight increase during regional transport. The contrast between hRH and lRH emphasize the importance of moisture on modifying the aerosol
compositions and hygroscopicity in the PBL.

The increase of $\kappa$ was in line with the increase of particle size, and both factors contributed to the increase of particle extinction due to particle hygroscopic growth. The enhancement of aerosol optical depth (AOD) due to water uptake ranged at 1.0-1.20 for PBL under lRH and LFT, but could reach as high as 4.4±1.3 in the PBL under hRH. The comparison of in-situ constrained AOD and AERONET AOD revealed that there
was about 80% and 18% of the AOD was contributed by aerosol hygroscopic growth under hRH and lRH respectively. Importantly, the most enhancement of $\kappa$ and extinction occurred at the top of PBL under wet condition, leading to enhanced positive vertical gradient of AOD distribution. This evolution process from the surface to the top of PBL should be considered, given the particle information on the surface may not represent that on top of the PBL where particle activation will mostly occur.
The results here showed the globally used $\kappa$=0.3 (Pringle et al., 2010) may be applied only when the anthropogenic emissions are after significant secondary processing, such as in this study $\kappa$ reached 0.34 at the top of PBL during high moisture condition, or above the PBL where regional transport advected aged pollutants. The fresher emissions, or the emissions after being scavenged, showed lower $\kappa$ at 0.20-0.25 as shown here. This study provides a frame of particle hygroscopicity under different pollution and
moisture level over this region influenced by intense anthropogenic activities. The increased $\kappa$ and particle size towards the top of PBL under high moisture condition tends to result in feedback effects, allowing enhanced water content containing in particle due to hygroscopic growth, and this will facilitate the aqueous reactions (Liu et al., 2018b) and lead to further radiative impacts.

**Data availability.** Processed data are available from the file sharing link





(https://pan.baidu.com/s/1P4B7Of_mbyJhBgvpD6zAMA&shfl=sharepset) using extracting code dhsq.

**Author contributions.** QL, DL, QG, PT, FW, DZ, KB, YW, SD, KH, and JZ were involved in collecting, processing and analysis of aircraft and ground data. QL and DL carried out the data analysis, with significant inputs from DD and CZ. QL and DL wrote the paper. QL and all authors contributed to the discussions.

**Competing interests.** The authors declare that they have no conflict of interest.

**Acknowledgments.** This work was supported by National Key Research and Development Program of China (No. 2016YFC0203302), the National Natural Science Foundation of China (No. 41975177, 41875044, 41875167), and the Beijing Natural Science Foundation (No. 8192021). Part of this work is supported by the National Center of Meteorology, Abu Dhabi, UAE under the UAE Research Program for Rain Enhancement Science.



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



Table 1. Flight time schedules and corresponding planetary boundary layer height and surface RH.

| Date | Flight time | PBL height | Surface RH |
|---|---|---|---|
| Nov. 13th | 09:40-12:00 | 1200m | 85% |
| Nov. 13th | 16:30-18:25 | 1000m | 82% |
| Nov. 15th | 10:00-12:40 | 1700m | 31% |
| Nov. 15th | 15:30-17:10 | 1600m | 19% |
| Nov. 16th | 10:25-12:20 | 1000m | 55% |
| Nov. 16th | 15:45-18:25 | 900m | 34% |
| Nov. 17th | 09:25-10:45 | 1200m | 65% |
| Nov. 17th | 15:35-17:10 | 1200m | 64% |
| Nov. 18th | 09:25-11:20 | 1100m | 77% |
| Dec. 16th | 12:30-16:05 | 500m | 29% |
| Dec. 17th | 12:40-16:10 | 500m | 32% |
| Dec. 18th | 12:10-14:30 | 350m | 40% |
| Dec. 19th | 12:25-16:20 | 350m | 37% |





Table 2. Density, hygroscopicity parameter ($\kappa$) and refractive indices of pure composition used in this study.

| Species | Density (kg m$^{-3}$) | $\kappa$ | Refractive index |
|---|---|---|---|
| $NH_4NO_3$ | 1725 | 0.68 | $1.6 - 0i$ |
| $(NH_4)_2SO_4$ | 1769 | 0.52 | $1.53 - 0i$ |
| $NH_4HSO_4$ | 1780 | 0.56 | $1.47 - 0i$ |
| SOA | 1400 | 0.1 | $1.46 - 0.021i$ |
| POA | 1000 | 0 | $1.46 - 0.021i$ |
| Black Carbon | 1800 | 0 | $1.85 - 0.79i$ |
| Water | 1000 | | $1.3+0i$ |

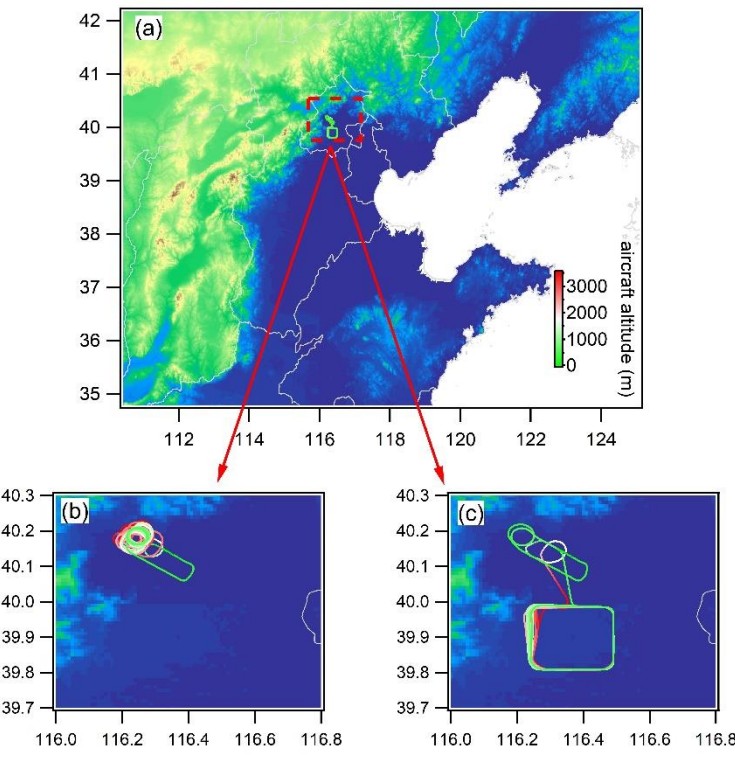

Figure 1. Flight tracks mapping on the terrain map. (a) the surrounding terrain, (b) flight tracks in
625  November 2016 (c) and in December 2016 respectively.





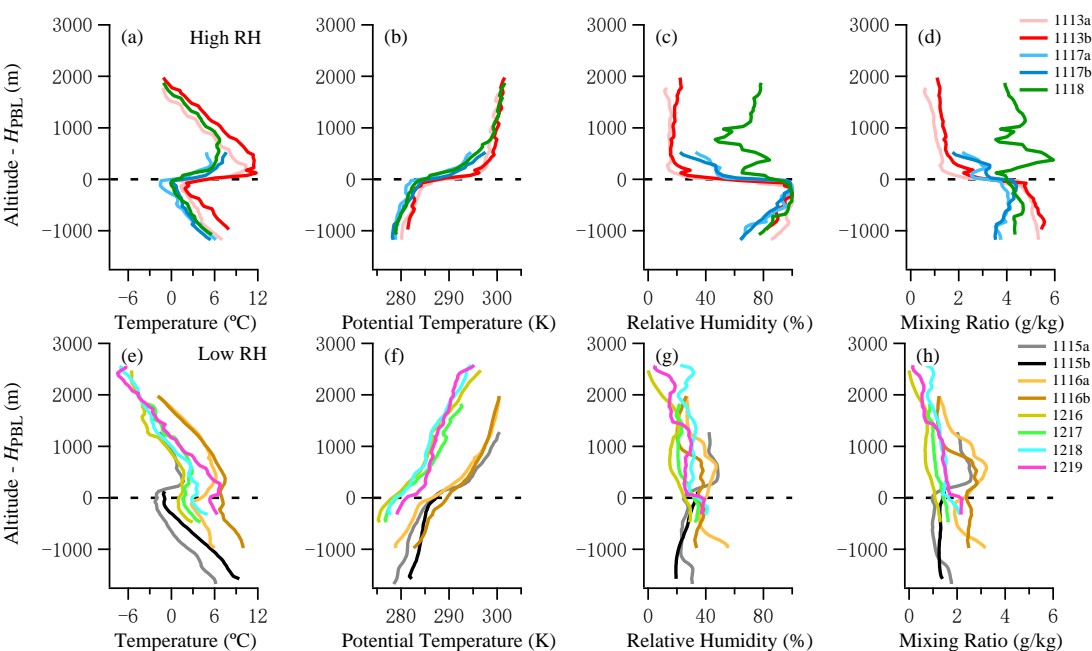

Figure 2. Vertical profiles of in-situ measured meteorological parameters under high RH and low RH conditions during the experiment, y-axis denotes the a.s.l. height relative to the planetary boundary layer height.





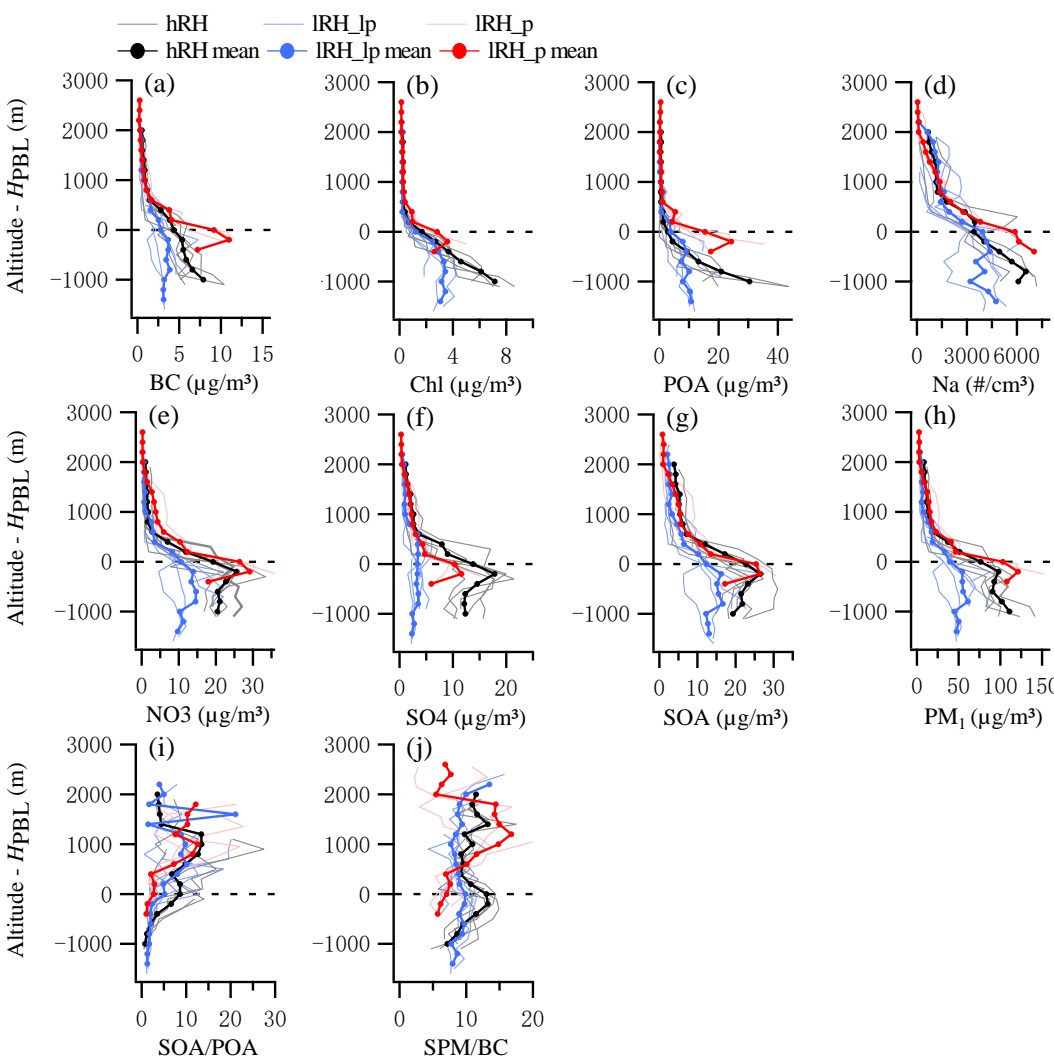

Figure 3. Vertical profiles of aerosol properties under lRH and hRH conditions. (a–d) primary aerosol
components and number concentrations, Na denotes the number concentration at 0.12-3 μm measured by
the PCASP, (e–h) secondary aerosol components and mass concentrations, (i) ratio of SOA over POA, (j)
ratio of SPM (secondary particulate matters) over BC. The solid lines show mean value in 100m altitude
bin.



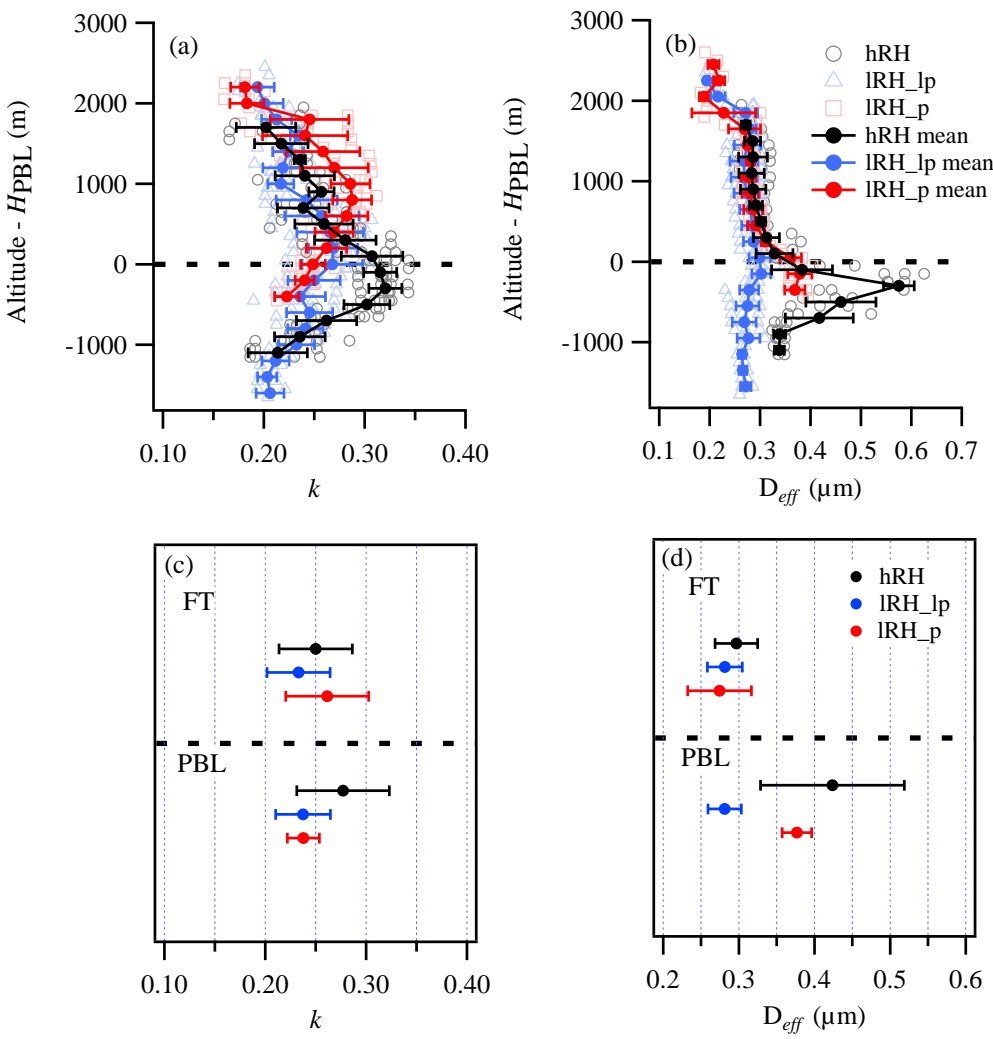

Figure 4. Vertical profiles of (a) hygroscopic parameter $\kappa$, (b) effective diameter $D_{eff}$ of dry particle, and (c-d) mean±σ in the LFT and PBL corresponding with the upper panel.





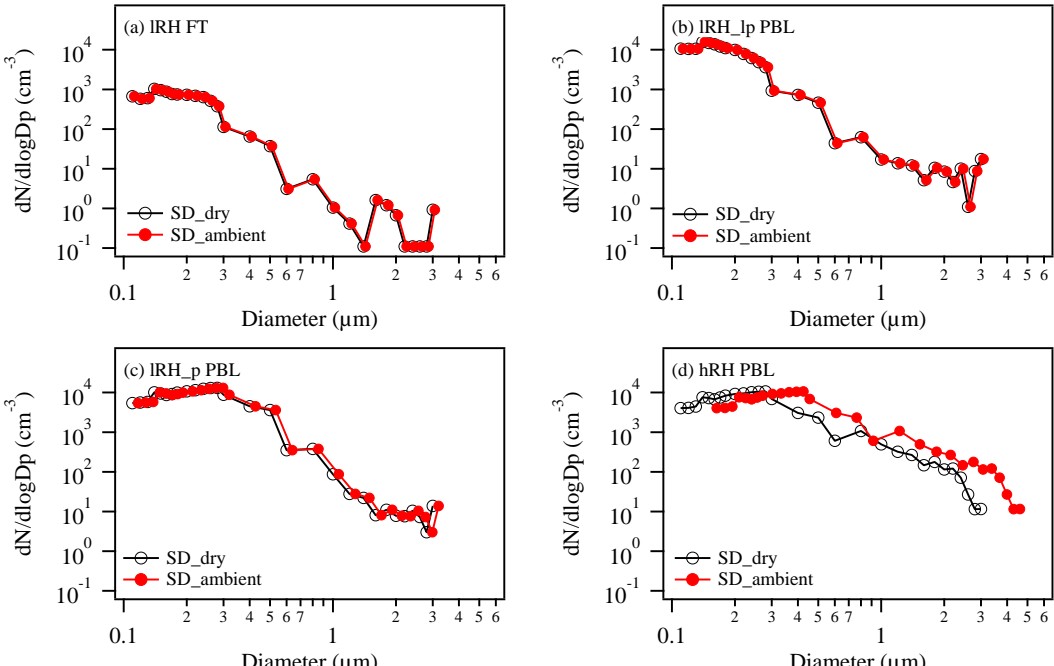

Figure 5. Measured dry size distribution and estimated ambient size distribution by considering the hygroscopic growth on aerosol, for (a) lower free troposphere under low RH (lRH FT), (b) PBL under low RH and less polluted condition (lRH_lp PBL), (c) PBL in the polluted but low RH condition (lRH_p PBL), (d) PBL under high RH condition (hRH PBL).

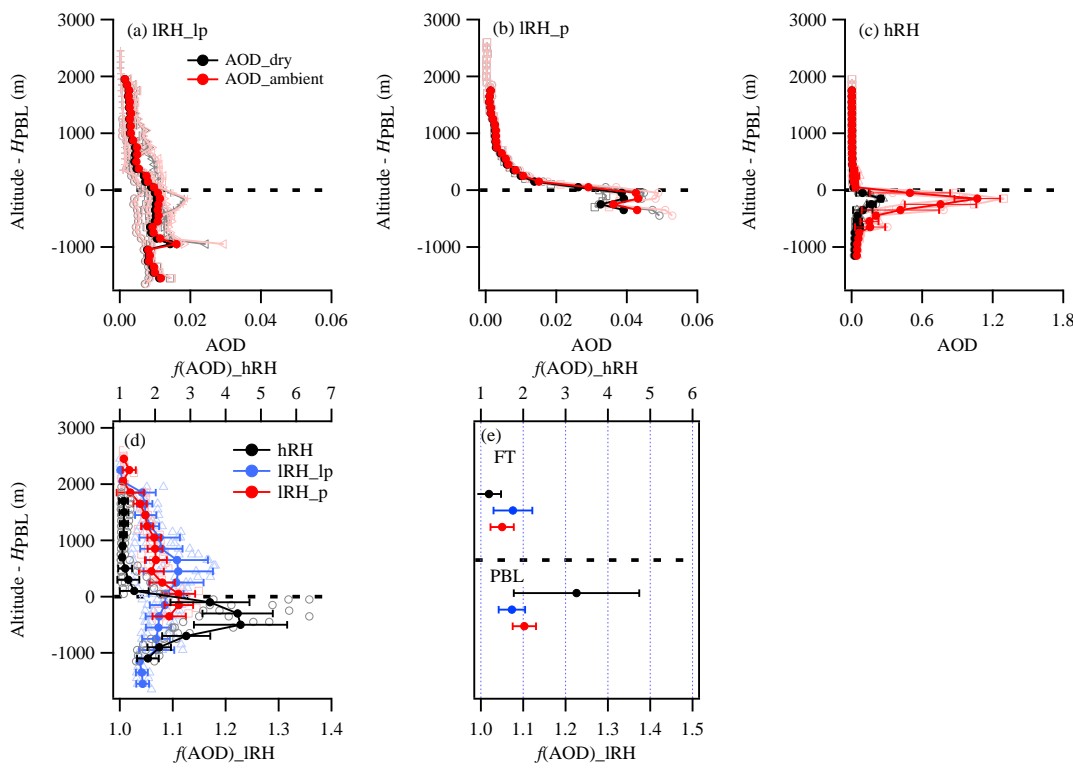

Figure 6. Vertical profiles of AOD under lRH and hRH conditions, (a) low RH and less polluted condition (lRH_lp), (b) low RH and polluted condition (lRH_p), (c) high RH condition (hRH). The grey and light red lines indicate the AOD for dry and ambient RH conditions respectively; (c-d) vertical profiles of $f$(AOD) (the ratio of calculated ambient AOD and dry AOD) and corresponding mean±σ in the FT and PBL. Note that the $f$(AOD) for hRH uses the top x-axis.





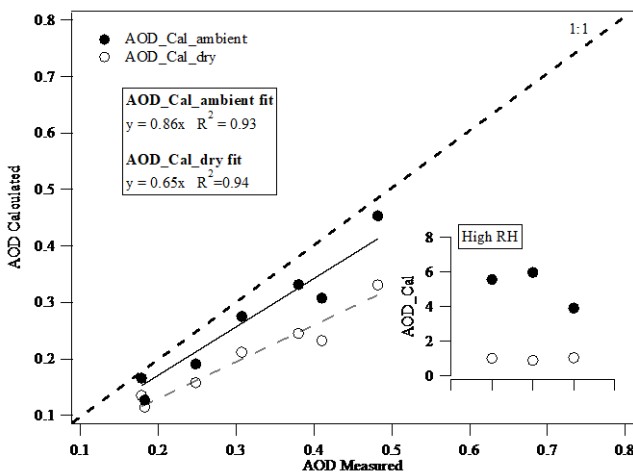

Figure 7. Comparison of in-situ measured dry AOD and ambient AOD with AERONET measurement.





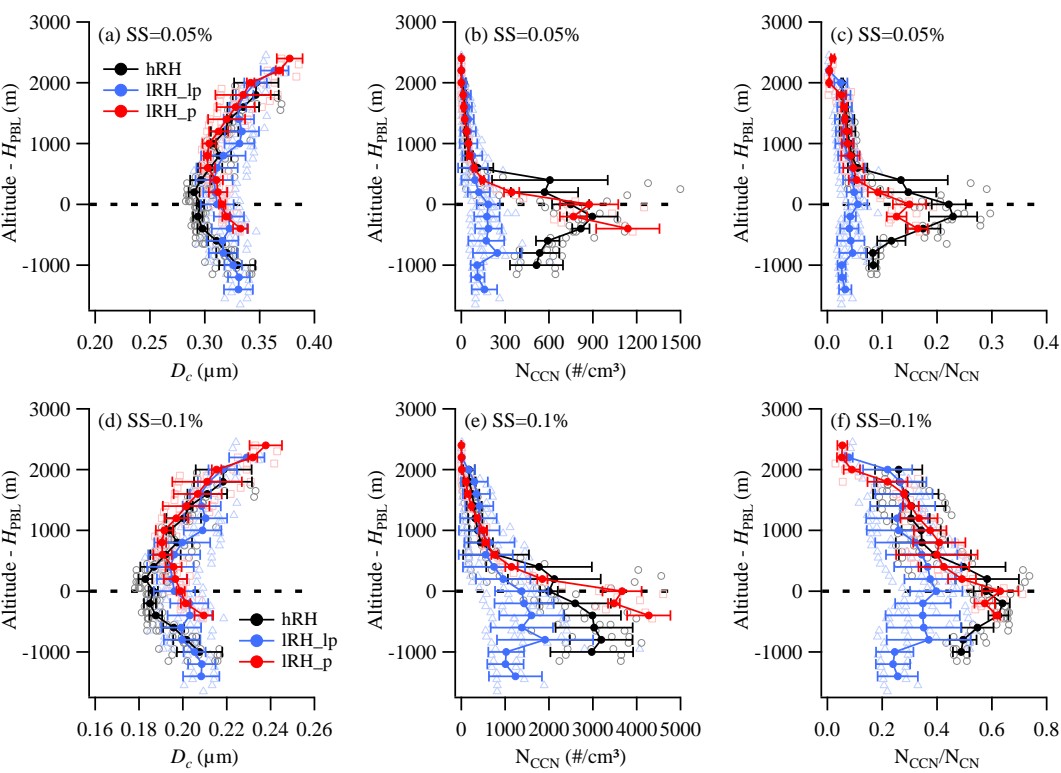

Figure 8. Vertical profiles of aerosol activation properties under lRH and hRH conditions, (a-c) critical diameter ($D_c$), number concentration of CCN ($N_{CCN}$) and the ratio of $N_{CCN}$ and $N_{CN}$ ($N_{CCN}/N_{CN}$) at supersaturation (SS) of 0.05%, using PCASP measured size distribution, (d-f) $D_{50}$, $N_{CCN}$, and $N_{CCN}/N_{CN}$ at SS=0.1%. The black, blue and red dot lines denote the profiles under hRH, lRH_lp, and lRH_p conditions, respectively.