# Peer review of "Vertical characteristics of aerosol hygroscopicity and impacts on optical properties over the North China Plain during winter"

_Atmospheric Chemistry and Physics, 2019_

## Referee Comment (RC1) · Anonymous Referee #1 · 1 Jan 2020

Hygroscopicity of aerosols is a key factor determining the direct and indirect climate effect of aerosols to some extent. The hygroscopicity also influences the chemical processes in the atmosphere and the development of PBL. Despite of many studies of hygroscopicity over last decades, very few of studies focused on the vertical distribution of this key parameter over Beijing region. This work presents a valuable direct observation of vertical profiles of hygroscopicity and analyses the corresponding impacts on optical aerosol depth and cloud droplet activation. The analysis is solid and the results are with interests to the community. I recommend this paper published in ACP with some minor revision.

[Figure]

Minor concerns:

1) In the 1st paragraph of introduction, authors correctly introduced the importance of hygroscopicity on the direct and indirect radiative forcing of aerosols. The importance of hygroscopicity is not only limited on these. Hygroscopic growth of aerosols can also directly influence the consistence of observations of aerosol mass and chemical compositions, leading to less robustness in the analyses of spatial/temporal variances and chemical mechanism studies (Chen et al., 2018). Therefore, the observation of hygroscopicity profile is critical for atmospheric science, with respect to scientific and measurement perspectives.

2) Earlier researches of the aerosol-PBL interaction worth the credit. For example, in lines 68-70, a previous comprehensive work, with combination of observations and modelling, nicely demonstrated the suppression effect of aerosol on the development of PBL and then enhance the pollution in Chinese megacities, including Beijing, Nanjing and etc. (Ding et al., 2016).

3) Equation 3. AMS also observes chloride, which is however not described in the Eq.3. How does chloride considered in the study? This could be important, because inorganic chloride usually present as highly hygroscopic.

4) Line 212. As described in the manuscript, AOD is calculated at a wavelength of 800nm. Is there a special reason of choosing 800 nm, instead of the conventional 550 nm, nor the 880 nm used of retrieve BC? And, for the AOD from AERONET, AOD at 800 nm is not directly available. Which wavelength of AOD from AERONET was used to compare against the aircraft observation-derived AOD should be described in the Method. Or, the method for converting AERONET AOD to a wavelength of 800 nm should be described in the Method.

5) Line 333. 'HGF increased from 1.2 to 2.1 by a factor of 1.8'. This description is corresponding to particles of what size?

6) This work uses kappa-Koehler theory to estimate the CCN number concentration. I just wonder that if authors also performed CCN observation on the aircraft. The comparison with direct CCN observations would be interesting to see.

7) line 430-431. 'but the results here show that the enhanced RH will increase both dry particle size and hygroscopicity through a variety of aqueous reactions and processes.'. This conclusion is not supported by the results of the presented study, we need cite other studies to support this statement here.

Technical corrections:

1) Line 33. Should be 'slightly increase'

2) Line 25-31. This sentence is too long and difficult to understand. Please rephrase it.

3) line 38. Please specify the meaning of 'boundary layer processing'.

4) line 259. Should be 'correctly'.

5) line 285-286. Effective diameter for dry or wet particles?

6) Line 348. Please specify here, it is the AOD of PBL or LFT?

7) line 436. 'The aerosols at higher level, which showed a smaller size and lower hygroscopicity,...'. As my understanding, this work seems show that hygroscopicity increase as altitude increases. Please check this statement.

8) Please give the full description of short-names before use them and in the caption of figures. This would improve the reading experience. For example, it is difficult to understand 'lRH_lp' in figure 4.

References:

Chen, Y., Wild, O., Wang, Y., Ran, L., Teich, M., Größ, J., Wang, L., Spindler, G., Herrmann, H., van Pinxteren, D., McFiggans, G., and Wiedensohler, A.: The influence of impactor size cut-off shift caused by hygroscopic growth on particulate matter loading and composition measurements, Atmospheric Environment, 195, 141-148, https://doi.org/10.1016/j.atmosenv.2018.09.049, 2018.

Ding, A. J., Huang, X., Nie, W., Sun, J. N., Kerminen, V.-M., Petäjä, T., Su, H., Cheng, Y. F., Yang, X.-Q., Wang, M. H., Chi, X. G., Wang, J. P., Virkkula, A., Guo, W. D., Yuan, J., Wang, S. Y., Zhang, R. J., Wu, Y. F., Song, Y., Zhu, T., Zilitinkevich, S., Kulmala, M., and Fu, C. B.: Enhanced haze pollution by black carbon in megacities in China, Geophysical Research Letters, 43, 2873-2879, doi:10.1002/2016GL067745, 2016.

---

## Referee Comment (RC2) · Anonymous Referee #2 · 13 Jan 2020

The hygroscopic properties of aerosol in the polluted East Asia has been long studied from ground, but direct characterization of the vertical profile is very limited. This study provides vertical profiles of particle hygroscopicity under different meteorological conditions, by considering both particle size and chemical composition as well as evaluating the hygroscopic growth on optical properties. It provides insights in evaluating the cause of pollution especially under high moisture condition. The manuscript is generally well written. I recommend publication in ACP after addressing the following comments:

1. This work has both size and chemical composition measurements. The important

message which could be delivered is the total CCN number concentration under certain SS%. It is better to highlight how much CCN could be in the highly polluted and less-polluted environment. 2. In the last section, discussion about CCN activation needs improvement, by including the discussions of CCN number concentration under different meteorological conditions. 3. It would be better to give some parameterizations of f(AOD) or f(RH).

Specific comments:

1. Show the location of AERONET site in Fig. 1. 2. In Table 1. the low PBLH corresponds with high RH? 3. Fig. 3 and Fig. 4, please describe the abbreviation of lRH, _P, _lp etc. in the caption. 4. Fig. 5, please provide the effective diameter in the figure. 5. The labels are too small for Fig. 7., the whole figure needs to be made larger. 6. line 316-318, how consistent with dry size? 7. Line 355-357, SS=1% can be deemed to be in convective system, a stratus may not reach as high as 1%, need to rewrite this part. 8. Line 409, framework. 9. the letter size in some figures are too small, please make them readable.

---

## Author Comment (AC1) · 7 Feb 2020

General comment:

Hygroscopicity of aerosols is a key factor determining the direct and indirect climate effect of aerosols to some extent. The hygroscopicity also influences the chemical processes in the atmosphere and the development of PBL. Despite of many studies of hygroscopicity over last decades, very few of studies focused on the vertical distribution of this key parameter over Beijing region. This work presents a valuable direct observation of vertical profiles of hygroscopicity and analyses the corresponding impacts on optical aerosol depth and cloud droplet activation. The analysis is solid and the results

are with interests to the community. I recommend this paper published in ACP with some minor revision.

[Response] We thank the referee for the positive comments and constructive suggestions, we have revised the manuscript according to the comments point by point.

Minor concerns

1) In the 1st paragraph of introduction, authors correctly introduced the importance of hygroscopicity on the direct and indirect radiative forcing of aerosols. The importance of hygroscopicity is not only limited on these. Hygroscopic growth of aerosols can also directly influence the consistence of observations of aerosol mass and chemical compositions, leading to less robustness in the analyses of spatial/temporal variances and chemical mechanism studies (Chen et al., 2018). Therefore, the observation of hygroscopicity profile is critical for atmospheric science, with respect to scientific and measurement perspectives.

[Response] We have added the related discussion in the revised version: "In addition, hygroscopic growth of aerosols can also directly influence the consistence of observations of aerosol mass and chemical compositions, leading to less robustness in the analyses of spatial/temporal variances and chemical mechanism studies (Chen et al., 2018). Therefore, the observation of hygroscopicity profile is critical for atmospheric science, with respect to scientific and measurement perspectives". Please see page 3, lines 47-51.

2) Earlier researches of the aerosol-PBL interaction worth the credit. For example, in lines 68-70, a previous comprehensive work, with combination of observations and modelling, nicely demonstrated the suppression effect of aerosol on the development of PBL and then enhance the pollution in Chinese megacities, including Beijing, Nanjing and etc. (Ding et al., 2016).

[Response] We have added the related discussion in the revised version. Please see

page 4, lines 74-76.

3) Equation 3. AMS also observes chloride, which is however not described in the Eq.3. How does chloride considered in the study? This could be important, because inorganic chloride usually present as highly hygroscopic.

[Response] Thanks for pointing this out. We have performed the analysis by including the composition of NH4Cl based on AMS measurement, and found he mean mass fraction of NH4Cl was only 3.6%±2.0% during the observations. sometimes the chloride mass concentrations were even lower than AMS detection limit, especially at high altitudes. Therefore, its contribution to bulk aerosol hygroscopicity could be ignored during the observations. We have added this discussion in the revised version. Please see page 8, lines 160-163.

"By including the ammonium chloride, a mass fraction of 3.6%±2.0% was found throughout the experiment, and chloride concentration was mostly lower than the lower AMS detection limit, thus its contribution to bulk aerosol hygroscopicity could be ignored during the observations."

4) Line 212. As described in the manuscript, AOD is calculated at a wavelength of 800nm. Is there a special reason of choosing 800 nm, instead of the conventional 550 nm, nor the 880 nm used of retrieve BC? And, for the AOD from AERONET, AOD at 800 nm is not directly available. Which wavelength of AOD from AERONET was used to compare against the aircraft observation-derived AOD should be described in the Method. Or, the method for converting AERONET AOD to a wavelength of 800 nm should be described in the Method.

[Response] We have double checked with our analysis and found we did choose the AERONET $\lambda$ and did the calculation at $\lambda$=870nm. We have now corrected in the revised version in line 194 and also marked this in the revised Figure 7.

5) Line 333. 'HGF increased from 1.2 to 2.1 by a factor of 1.8'. This description is

corresponding to particles of what size?

[Response] The HGF as a function of RH is calculated at diameter 200nm, and is independent of particle size (within 1%) at submicron diameter. We have added this information in the revised version.

6) This work uses kappa-Koehler theory to estimate the CCN number concentration. I just wonder that if authors also performed CCN observation on the aircraft. The comparison with direct CCN observations would be interesting to see.

[Response] Thanks for pointing this out. We admit that it is very important to compare our estimated result with direct measurement. However, there was no direct CCN observation on the aircraft., our discussion thus mainly focused on the difference of CCN activation among different supersaturation conditions.

7) line 430-431. 'but the results here show that the enhanced RH will increase both dry particle size and hygroscopicity through a variety of aqueous reactions and processes.'. This conclusion is not supported by the results of the presented study, we need cite other studies to support this statement here.

[Response] We have added a few references to support this statement as referee suggests.

Technical corrections:

1) Line 33. Should be 'slightly increase'

[Response] Revised.

2) Line 25-31. This sentence is too long and difficult to understand. Please rephrase it.

[Response] Revised. Please see lines 27-32.

3) line 38. Please specify the meaning of 'boundary layer processing'.

[Response] Revised. Please see lines 37-39.

"These results emphasize the important evolution of aerosol water-uptake capacity in the PBL, especially under high RH condition."

4) line 259. Should be 'correctly'.

[Response] Revised.

5) line 285-286. Effective diameter for dry or wet particles?

[Response] It is for dry particle size. We have clarified this in the revised version. Please see line 258.

6) Line 348. Please specify here, it is the AOD of PBL or LFT?

[Response] It is the AOD integrated over the PBL. We have clarified this in the revised manuscript, please see line 310.

7) line 436. 'The aerosols at higher level, which showed a smaller size and lower hygroscopicity,...'. As my understanding, this work seems show that hygroscopicity increase as altitude increases. Please check this statement.

[Response] We have clarified this point as: "The $\kappa$ increased with altitude in the PBL and showed a maxima at the top of PBL, then decreasing with altitude in the LFT". Please see lines 273-274.

8) Please give the full description of short-names before use them and in the caption of figures. This would improve the reading experience. For example, it is difficult to understand 'lRH_lp' in figure 4.

[Response] We added the description of the short-names in the caption of revised Figure 3 and Figure 4.

---

## Author Comment (AC2) · 7 Feb 2020

General comment:

The hygroscopic properties of aerosol in the polluted East Asia has been long studied from ground, but direct characterization of the vertical profile is very limited. This study provides vertical profiles of particle hygroscopicity under different meteorological conditions, by considering both particle size and chemical composition as well as evaluating the hygroscopic growth on optical properties. It provides insights in evaluating the cause of pollution especially under high moisture condition. The manuscript is generally well written. I recommend publication in ACP after addressing the following

comments:

[Response] We thank the positive comments from the referee and we have revised modified the manuscript according to the comments point by point.

1. This work has both size and chemical composition measurements. The important message which could be delivered is the total CCN number concentration under certain SS%. It is better to highlight how much CCN could be in the highly polluted and less polluted environment.

[Response] We have added the description in lines 373-375 and give the total CCN number concentration in the highly polluted and less polluted environment as below: "The total CCN number concentration present a distinct difference between clean and polluted environment. For example, in the PBL, the averaged CCN number concentration at SS=0.05% was only 167±44 cm-3 under lRH_lp period, and increased to 765±199 cm-3 under highly polluted environment, e.g., lRH_p and hRH conditions."

2. In the last section, discussion about CCN activation needs improvement, by including the discussions of CCN number concentration under different meteorological conditions.

[Response] We added the description of CCN number concentration under different meteorological conditions as below, please see lines 376-377: "For SS=0.1%, the averaged CCN number concentration increased to 1370±297 cm-3, 3807±415 cm-3, and 2797±438 cm-3 under lRH_lp, lRH_p, and hRH conditions respectively."

3. It would be better to give some parameterizations of f(AOD) or f(RH).

[Response] We added the statement of "During the observations, f(AOD) in the PBL increased with altitude at 0.03 0.09, 2.43 per km elevation under lRH_lp, lRH_p, and hRH conditions respectively". Please see lines 318-320.

Specific comments: 1. Show the location of AERONET site in Fig. 1.

[Response] We have added the location of Peking University AERONET site in revised Figure 1, which supplied the measured AOD data in this study.

2. In Table 1. the low PBLH corresponds with high RH?

[Response] No obvious correlation between the PBLH and RH during the observations. For example, the PBLH in Dec. 17th was 500m and the surface RH was only 32%, however the PBLH in Nov. 13th reached to 1200m with a high surface RH of 85%.

3. Fig. 3 and Fig. 4, please describe the abbreviation of lRH, _P, _lp etc. in the caption.

[Response] Thanks for pointing this out. We added the description of the abbreviation in the caption of Figure 3 and Figure 4.

4. Fig. 5, please provide the effective diameter in the figure.

[Response] We add the effective diameter under different conditions in Figure 5.

5. The labels are too small for Fig. 7., the whole figure needs to be made larger.

[Response] We adjusted the Figure 7.

6. line 316-318, how consistent with dry size?

[Response] The vertical profiles of dry aerosol effective diameter (Deff) are shown in Figure 4b. As it shown, the Deff increased with altitude in the PBL for all conditions, which were consistent with the variations of aerosol hygroscopic parameter ($\kappa$). We revised the related discussion, please see lines 326-327.

7. Line 355-357, SS=1% can be deemed to be in convective system, a stratus may not reach as high as 1%, need to rewrite this part.

[Response] We note that the statement of "the supersaturation (SS) for status clouds in clean condition often exceeded 1%" is not accurate. We have rechecked some of the literatures that the max SS for stratus clouds over polluted continental regions usually be slightly less than 0.1% during wintertime (Hudson and Noble, 2014; Hudson et al.,

2010). We thus discussed the CCN activity at SS=0.05% and 0.1% respectively. We have revised that statement.

8. Line 409, framework.

[Response] Revised.

9. the letter size in some figures are too small, please make them readable

[Response] Revised.
* * *